# Chimeric Virus-like Particles Co-Displaying Hemagglutinin Stem and the C-Terminal Fragment of DnaK Confer Heterologous Influenza Protection in Mice

**DOI:** 10.3390/v14102109

**Published:** 2022-09-23

**Authors:** Cui-Cui Liu, De-Jian Liu, Xin-Yu Yue, Xiu-Qin Zhong, Xuan Wu, Hai-Yan Chang, Bao-Zhong Wang, Mu-Yang Wan, Lei Deng

**Affiliations:** 1Hunan Provincial Key Laboratory of Medical Virology, College of Biology, Hunan University, Changsha 410082, China; 2College of Life Sciences, Hunan Normal University, Changsha 410082, China; 3Center for Inflammation, Immunity & Infection, Georgia State University, Atlanta, GA 30303, USA; 4Beijing Weimiao Biotechnology Co., Ltd., Haidian District, Beijing 100080, China

**Keywords:** influenza virus, hemagglutinin stem, virus-like particle, DnaK, universal vaccine

## Abstract

Influenza virus hemagglutinin (HA) stem is currently regarded as an extremely promising immunogen for designing universal influenza vaccines. The appropriate antigen-presenting vaccine vector would be conducive to increasing the immunogenicity of the HA stem antigen. In this study, we generated chimeric virus-like particles (cVLPs) co-displaying the truncated C-terminal of DnaK from *Escherichia coli* and H1 stem or full-length H1 antigen using the baculovirus expression system. Transmission electronic micrography revealed the expression and presentation of H1 stem antigens on the surface of VLPs. Vaccinations of mice with the H1 stem cVLPs induced H1-specific immune responses and provided heterologous immune protection in vivo, which was more effective than vaccinations with VLPs displaying H1 stem alone in protecting mice against weight loss as well as increasing survival rates after lethal influenza viral challenge. The results indicate that the incorporation of the truncated C-terminal of DnaK as an adjuvant protein into the cVLPs significantly enhances the H1-specific immunity and immune protection. We have explicitly identified the VLP platform as an effective way of expressing HA stem antigen and revealed that chimeric VLP is an vaccine vector for developing HA stem-based universal influenza vaccines.

## 1. Introduction

Seasonal influenza is an acute respiratory disease that can cause significant morbidity and mortality worldwide. The past global influenza pandemic outbreaks imposed severe threats to human public health and societal order. Vaccination is considered the most effective way of preventing influenza infection. Licensed influenza vaccines are largely based on inducing neutralizing antibodies against the HA head domain, thereby providing effective protection against antigenic matching influenza strains [1]. Due to the lack of preformed adaptive immunity, even vaccinees immunized with seasonal influenza vaccines are still susceptible to the genetically reassorted influenza A viruses [2]. The severity of influenza diseases combined with the unpredictable protection efficacy of conventional vaccines highlights the necessity and urgency of developing universal influenza vaccines that can induce broad and effective protection against a wide range of influenza viruses.

The homo-trimeric HA is the most abundant glycoprotein on the surface of influenza A and B viruses and is regarded as the target antigen to develop influenza vaccines conferring neutralizing immunity [3]. The HA globular head domain contains a receptor binding site specific to sialic acids that are typically located at the terminus of host glycan chains [4]. The extensive receptor binding by multiple HAs on an influenza virion would lead to irreversible attachment and initiate endocytosis, which would in principle be blocked by HA-specific antibodies targeting the receptor-binding site [5]. However, the hypermutation of the HA head domain enables the influenza virus progeny to escape from recognition by these antibodies [6]. In recent years, an increasing number of cross-reactive neutralizing monoclonal antibodies have been identified, and their corresponding epitopes are distributed in almost the entire stem region [7,8,9,10,11], implying that the HA stem epitopes are relatively well conserved and would potentially elicit broadly neutralizing immune protection. Modified HA stem immunogens stabilized by an exogenous trimerization motif or structurally supported by solid-geometry-compatible self-assembling protein nanoparticles proved to retain the native conformation of HA stem and efficiently induce HA stem-specific neutralizing immunity in vivo [12,13]. Another approach by sequential immunizations of laboratory animals with chimeric HAs that are composed of stem regions from the same HA and head regions from divergent influenza subtypes efficiently induced stem-specific cross-neutralizing antibodies and heterologous protection [14]. Furthermore, a recent clinical report demonstrated that immunizations with inactivated influenza vaccines containing such HA chimeras significantly boosted broadly neutralizing immunity in vaccinees with preexisting stem-specific antibody responses [15]. 

Influenza VLP morphologically and biochemically resembles intact virions while lacking viral genetic materials and is recognized as an efficient and safe platform for vaccine development. The repetitive and highly organized molecular array of antigens displayed on the VLP surface enables the activation of B lymphocyte receptors and potentially increases antigen-specific immune responses. However, if an immune-subdominant domain, such as the HA stem, was not ready to access, B cell receptors targeting the epitopes in this region would neither interact with high avidity nor cluster to develop a strong antibody response. Therefore, we hypothesized that introducing ample space intervals in between the membrane-anchored H1 molecules on the VLP surface by co-displaying another type of relatively shorter molecules would potentially increase the chance of B cell receptor recognition in the membrane-proximal stem region and thus enhance HA stem-specific immune responses. The adjuvant protein with an appropriate size would serve as a rational option for co-displaying on the cVLPs. Ideally, the second type of protein not only serves as a paling to sterically separate the closely arrayed H1 molecules but also exerts adjuvant effects on enhancing specific immunity.

DnaK is the canonical heat-shock protein 70 chaperone family member, which is a stress-inducible and ubiquitous protein that maintains proteomic homeostasis [16]. *Escherichia coli* (*E. coli*) DnaK comprises a 44-kDa N-terminal ATPase domain, a 25-kDa C-terminal substrate-binding domain, and a dynamic random coil in between [17]. The previous investigation reported that the recombinant DnaK induced the production of proinflammatory cytokines interleukin (IL)-6 and tumor necrosis factor-α in mouse peritoneal macrophages via Toll-like receptor 4 [18]. Further studies revealed that the C-terminal peptide-binding domain rather than the N-terminal ATPase domain was able to drive the stimulation of human monocytes to produce cytokines and chemokines and the maturation of dendritic cells in vitro [19]. Vaccination experiments have revealed that DnaK conjugated to antigens could exert strong adjuvant effects [20]. Furthermore, bacterial DnaK was also found to be immunogenic in mice [21], Here, we reported a structure-guided design of influenza VLPs co-displaying the H1 stem and the C-terminal fragment of DnaK. Results showed that all the mice immunized with this immunogen were protected from heterologous influenza virus challenge. We have demonstrated that the cVLP is a viable platform for expressing the HA stem domain and co-displaying an additional protein adjuvant to effectively enhance vaccine immunogenicity.

## 2. Materials and Methods

### 2.1. Ethics Statement

All experiments associated with influenza viruses were performed in a biosafety level 2 laboratory at Hunan University. All animal experiments were strictly carried out in accordance with the ethical guidelines for animal experiments at Hunan University. The mice challenge experiments were performed in the biosafety level 2 animal facility at Hunan Normal University. 

### 2.2. Cell Lines and Influenza Viruses

Spodoptera frugiperda 9 (Sf9) insect cells were cultured in serum-free SF900 II medium in spinner flasks (Suzhou world-medium Biotechnology Co, Ltd., Suzhou, China) at 27 °C at a speed of 110 rpm. Madin Darby Canine Kidney (MDCK) cells or human lung carcinoma cell line A549 cells were grown and maintained in Dulbecco’s modified Eagle medium (DMEM) (Gibco) supplemented with 10% fetal bovine serum (FBS) at 37 °C in 5% CO_2_ condition. Mouse-adapted influenza H1N1 A/New Jersey/8/1976 (NJ76) and H1H1 A/Puerto Rico/8/1934 (PR8) viruses used in this study were prepared in the method described before [22]. 

### 2.3. Construction and Expression of the Chimeric VLPs 

To increase the integration efficacy of the displayed antigen on the chimeric VLPs, we constructed the recombinant pFastbac-Dual plasmid by inserting M1 (GenBank Protein Accession: NP_040978.1) and H1 stem (the sequence is modified from the influenza H1N1 strain PR8) or full-length H1 (the sequence is from the influenza H1N1 strain A/New Caledonia/20/1999, NC99, GenBank Protein Accession: AFO65027.1) encoding genes into two independent open reading frames under the controls by baculovirus polyhedrin promoter and p10 promoter, respectively. The PR8 H1 head domain (amino acids 53–320) of HA1 was replaced with a flexible linker GGGG, and these site-mutations I61Y, F63T, V66T, and L73Y were introduced into the HA2 B loop. The signal peptide sequence at the 5’ end of H1 was replaced by the melittin signal peptide to enhance the protein expression. 

The nucleotide sequence encoding the tDnaK (GenBank Protein Accession: MCJ3063044.1, amino acids 385–610; PDB: 1DKZ) with the melittin signal peptide at the N-terminal and the H1 transmembrane and cytosolic domains at the C-terminal was inserted into the open reading frame in the pFastbac-Dual plasmid for the expression of tDnaK in the baculovirus-infected Sf9 cells. The amino acid sequence of tDnaK is identical to the DnaK from *E. coli*, *Salmonella enterica*, and *Shigella sonnei*. The protein structures of the PR8 H1 stem and tDnaK were predicted by using the I-TASSER server (http://zhanglab.dcmb.med.umich.edu; accessed on 22 May 2022 and 26 June 2022 for the structure generations of PR8 H1 and tDnaK, respectively).

Recombinant baculoviruses were generated by transfection with the recombinant bacmids into Sf9 cells. The cVLPs presenting the full-length H1/H1 stem were produced from Sf9 cells infected with the recombinant baculovirus expressing M1 and the full-length H1/H1 stem. The cVLPs co-presenting the full-length H1/H1 stem and the tDnaK proteins were produced from Sf9 cells co-infected with the recombinant baculovirus expressing M1 and full-length H1/H1 stem and the recombinant baculovirus expressing the tDnaK. Prior to VLP expression and purification, the recombinant baculoviruses produced from the bacmid-transfected Sf9 cells were passaged twice for amplifying virus titers. Then, 0.5 mL baculoviruses inoculum from the second-round passages was added to 400 mL Sf9 cell suspension culture for infection. Four days post-infection, Sf9 cell culture supernatants were harvested and then clarified by centrifugation at 8000× *g* for 15 min at 4 °C to remove Sf9 cells and cellular debris. The virion particles were pelleted by ultracentrifugation at 130,000× *g* for 1 h at 4 °C and further purified using a 15–30–45–60% discontinuous sucrose gradient ultracentrifugation at 130,000× *g* for 1 h at 4 °C. The layer containing the chimeric VLPs was collected and resuspended in sterile PBS. The protein concentration of VLP samples was determined by using a BCA assay (TransGen Biotech, Beijing, China). The presence of the full-length H1, H1 stem, M1, and tDnaK were determined in Western blotting by using H1-specific serum from H1 (NC99) immunized mice, M1-specific monoclonal antibody C111 (Sino Biological, Beijing, China), and DnaK-specific polyclonal antibodies (CUSABIO, Wuhan, China).

### 2.4. Transmission Electron Microscopy

The transmission electron micrography was implemented by the Servicebio microscope facility (Wuhan, China). Five-microliter droplets of VLP suspension were adsorbed onto a 150-mesh copper grid for 1 min, followed by the removal of the remaining liquid with sterile absorbent paper. After the adsorption of 2% phosphotungstic acid for 30 s, the remaining liquid was removed. Then, the adsorption of phosphotungstic acid was repeated. Finally, VLPs were observed using a transmission electron microscope (HT7800, Hitachi) at the voltage of 80.0 kV.

### 2.5. In Vitro Assays for Inflammation Responses

Human A549 cells were plated at 10^5^ cells/mL in 6-well plates in the complete culture medium containing DMEM with 10% fetal bovine serum (FBS, BI, Kibbutz Beit Haemek, Israel). After treatment of 2 mL 10 µg/mL NC99 H1 VLP and NC99 H1-tDnaK VLP in the cell cultures, these plates were incubated at 37 °C and 5% CO_2_ for in vitro measurement of the expression levels of IL-1β and IL-6. Untreated cell cultures are the negative control. 

The cells were harvested at 24 h after treatment, then lysed using Trizol reagent (Sigma, St. Louis, MO, USA) and frozen at −20 °C for 4 h. After the cell samples were thawed on ice, the total RNA from each cell sample was extracted and was immediately reverse-transcribed into cDNA using reverse transcription and a cDNA synthesis kit (Takara Bio, Shiga, Japan). The cDNA samples were used as the templates to amplify the target genes by using qPCR (B21202, Bimake, Suzhou, China). The used primers were reported previously [23,24]. The abundances of transcripts were measured in four independent assays and normalized to the GAPDH mRNA level. Primers used in qPCR are listed in Table 1.

### 2.6. Animal Immunization and Challenge Studies

Six- to eight-week-old specific-pathogen-free (SPF) female BALB/c mice were purchased from Hunan SJA Laboratory Animal Co., Ltd. (Hunan, China). Five mouse groups (*n* = 5) were intraperitoneally immunized with 50 μL vaccines containing 25 μg VLPs adjuvanted with the incomplete Freund’s adjuvant (Sigma, St. Louis, MO, USA). Mice were immunized three times at 3-week intervals. Blood samples were collected by the submandibular vein puncture two weeks after each immunization. The blood samples were incubated at 37 °C for half an hour and then centrifuged at a speed of 3000× *g* at 4 °C for 10 min to collect the immune sera. The serum samples were stored at −20 °C for antibody titration. 

For virus challenge experiments, three weeks after the last boost immunization, mice were lightly anesthetized with 1% sodium pentobarbital and intranasally infected with 3 × LD50 of NJ76 viruses or PR8 viruses in 50 μL PBS. The body weight changes and survival rates of all groups were monitored daily for 14 days after infection. Weight loss of ≥20% or ≥25% was used as the endpoint at which mice were terminated. 

### 2.7. Determination of Specific IgG Antibody Titers

Levels of influenza virus-specific antibody titers of total IgG, IgG1, and IgG2a isotypes in serum were determined by using an enzyme-linked immunosorbent assay (ELISA). In brief, the 96-well plates were coated with 200 ng/well of inactivated PR8 or NJ76 H1N1 virus in coating buffer (0.1 M carbonate, pH = 7.5) at 4 °C overnight, then blocked with 5% skim milk powder in PBS for 2 h at 37 °C. Serially diluted serum samples were added to wells and incubated for 2 h at room temperature. After three washes with PBST comprising sterile PBS and 0.1% Tween20, the plates were incubated with HRP-conjugated secondary goat antibodies specific to mouse IgG, IgG1, or IgG2a for 1 h at room temperature. After five washes with PBST, TMB solution was added to develop a blue color, and the reaction was stopped by adding 2 M H_2_SO_4_. The absorbance at OD 40 nm was detected by using the microplate reader (DLJ-200, DLJ Bio-Tech, Jiangsu, China). 

### 2.8. Determination of Lung Viral Load

Immunized mice (*n* = 3 per group) were sacrificed on the 4th day post 1 × LD_50_ NJ76 H1N1 infection. Lungs were excised and homogenized in a 10% (*w*:*v*) serum-free DMEM medium. Lung homogenates were cleared from debris by centrifugation for 10 min at 10,000× *g*. MDCK cells were seeded in triplicate at 2 × 10^4^ cells per well in a 96-well plate and infected with 50 μL of a ten-fold series of diluted cleared lung homogenates. After 1 h incubation at 37 °C, 5% CO_2_, the lung homogenate dilutions were replaced with 100 μL serum-free medium supplemented with 100 μg/mL streptomycin, 100 U/mL penicillin (Solarbio, Beijing, China), 0.1 mM MEM non-essential amino acid (Gibco, Carlsbad, CA, USA), and 2 μg/mL TPCK-treated trypsin. After five days of incubation at 37 °C, 5% CO_2_, the presence of virus in the supernatant was assayed by measuring the hemagglutination activity in the supernatant, and virus titers were calculated by TCID_50_ using the method of Reed and Muench for calculation [25]. 

### 2.9. ELISPOT Assay

On the 4th-day post sublethal-dose infections with NJ76 H1N1, spleens of three mice in each group were collected and homogenized for splenocyte isolation. To evaluate the IFN-γ- and IL-4-secreting splenocytes, the capture antibody-precoated multiscreen 96-well filtration plates (BD Biosciences, Miami, FL, USA, or Dakewe, Biotech Co., Ltd., Shenzhen, China) were blocked with RPMI 1640 medium with 10% FBS, before the addition of 100 μL freshly prepared splenocyte suspensions at 2 × 10^7^ cells/mL in complete RPMI medium. After overnight stimulation with 50 μL, 10 μg/mL inactivated NJ76 virion solution at 37 °C, plates were overlaid with 50 μL, 2 μg/mL of biotinylated anti-mouse IFN-γ (BD Biosciences, USA) or anti-mouse IL-4 antibody (Dakewe, Biotech Co., Ltd., Shenzhen, China) and incubated for 1 h at 37 °C. After washing three times with PBST, 50 μL HRP-conjugated streptavidin (1:1000 in PBST) was added, and samples were incubated at room temperature for 1 h. After washing with PBST, 3-3′-diaminobenzidine tetrahydrochloride was added to develop spots in the plates. The plates were rinsed with tapping water and air-dried before counting and imaging using the stereomicroscope.

### 2.10. Serum Microneutralization Assay

Microneutralization assay was carried out as previously described [26]. Briefly, serum samples were heat-inactivated at 56 °C for 30 min, then two-fold serially diluted in the mixtures with final concentrations of 150 × TCID_50_ NJ76 H1N1 virus. The virus-serum mixtures were incubated at room temperature for 2 h, then added to MDCK monolayer cell cultures in the serum-free DMEM medium and incubated at 37 °C, 5% CO_2_ for 1 h. The culture was replaced with the medium containing 2 μg/mL TPCK-trypsin, 100 U/mL penicillin, and 100 μg/mL streptomycin and incubated at 37 °C, 5% CO_2_ for 72 h. Standard hemagglutination assays using 1% chicken red blood cells were performed to measure virus inhibition.

### 2.11. Statistical Analysis

T-test analysis was performed using GraphPad Prism 8. *P*-values less than or equal to 0.05 were considered significant. *, **, and **** represent *p*-value < 0.05, *p*-value < 0.01 and *p*-value < 0.001, respectively.

## 3. Results

### 3.1. Construction and Characterization of Influenza Chimeric VLPs

We designed four types of influenza cVLPs, each displaying the full-length H1 or the H1 stem on the membrane surface with or without co-displaying the C-terminal fragment of DnaK (Figure 1a). This truncated DnaK (designated as tDnaK) polypeptide spans the amino acid sequence from 385 to 610. The full-length H1, H1 stem, and tDnaK were fused with the melittin signal peptide at their N-terminal for the expression in insect cells and fused with the H1 transmembrane domain and cytosolic tail at their C-termini for protein-anchoring on the VLP membrane. To generate the recombinant H1 stem from PR8 H1, the amino acid sequence from 53 to 320 of HA1 in the head domain was replaced with a flexible linker GGGG, and the site-mutations I61Y, F63T, V66T, and L73Y were introduced into the HA2 B loop (Figure 1b). These cVLPs were produced from Sf9 insect cells that were infected or co-infected with recombinant baculovirus expressing M1 and the full-length H1/H1 stem and/or recombinant baculoviruses expressing tDnaK (Figure 1c).

**Figure 1 viruses-14-02109-f001:**
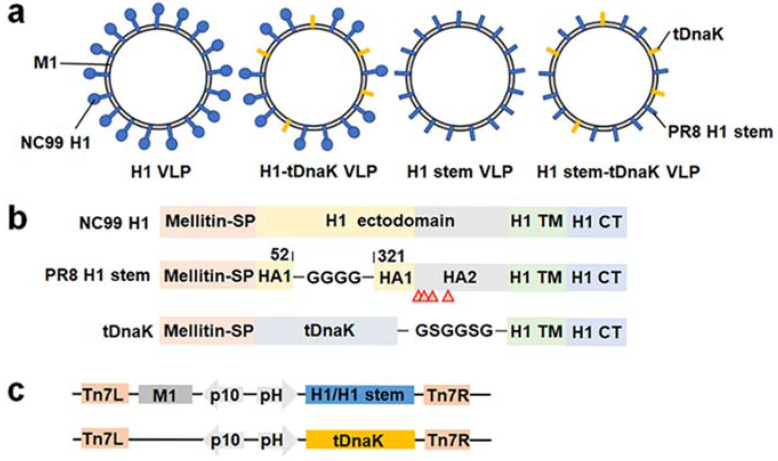
The construction of the cVLPs. (**a**) Schematic diagrams of the chimeric H1 VLP, H1-tDnaK VLP, H1 stem VLP, and H1 stem-tDnaK VLP. (**b**) Schematic diagrams of the designs of the full-length NC99 H1, PR8 H1 stem, and membrane-anchored tDnaK. Red arrows indicate site-mutations, I61Y, F63T, V66T, and L73Y in PR8 HA2. (**c**) Schematic diagrams of the designs of recombinant pFastBac Dual plasmids encoding M1 and H1/H1 stem, or membrane-anchored truncated DnaK alone. These exogenous genes were cloned into two independent open reading frames under the controls of the baculovirus polyhedrin promoter (pH) and the p10 promoter, respectively. All the cVLPs were purified by using density-gradient ultra-centrifugation. Western blotting analysis using recombinant H1 (NC99 H1N1)-specific mouse sera, the M1-specific monoclonal antibody C111, and DnaK-specific polyclonal antibodies clearly showed the presence of H1 and M1 in the H1 VLPs, H1, M1, and tDnaK in the H1-tDnaK VLPs, the H1 stem and M1 in the H1 stem VLPs, the H1 stem, M1 and tDnaK in the H1 stem-tDnaK VLPs (Figure 2a). To evaluate the size and conformation of H1 stem and tDnaK recombinant proteins, the structures of these two proteins were successfully predicted using the I-TASSER server (Figure 2b,c). Transmission electron micrography clearly showed the cVLP of approximately 80 nm in size and the good presentation of H1 stem antigens on the VLP membrane (Figure 2d). A cartoon image of the intercepted H1 stem (PDB: 1RU7) generated by using PyMOL was overlain on a spike in the TEM image with the scale bar of the same size and was found to be quite similar in size and shape to the spike on the VLP surface. According to the image superimposition analysis, we speculate that the intact fusion peptide and H1 transmembrane domain immobilize the H1 stem pre-fusion trimeric state.

**Figure 2 viruses-14-02109-f002:**
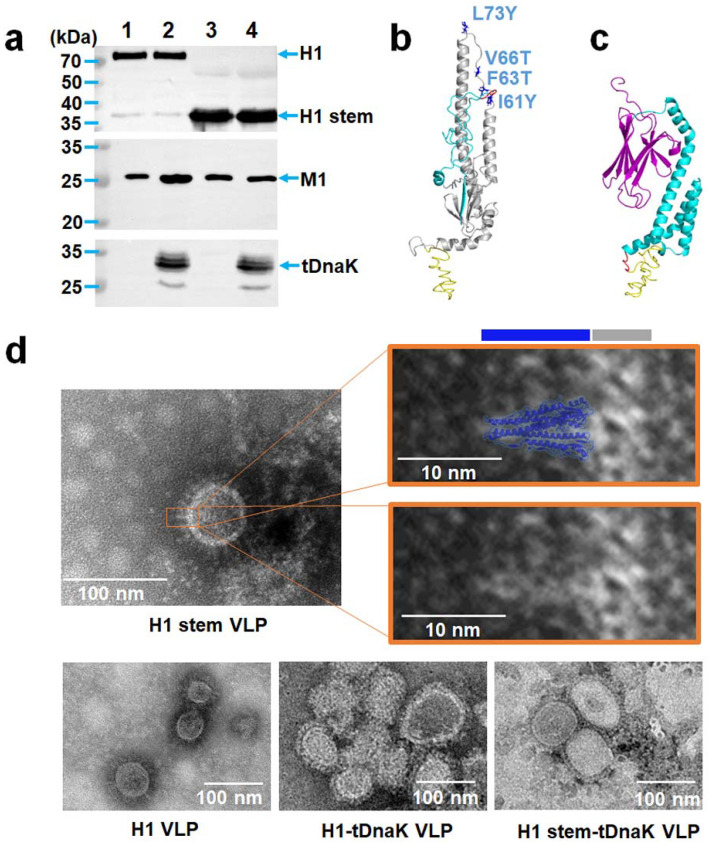
The characterization of the cVLPs. (**a**) Western blot analysis of the expressions of the full-length H1, H1 stem, M1, and tDnaK from the cVLPs, including (1) H1 VLPs, (2) H1-tDnaK VLPs, (3) H1 stem VLPs, and (4) H1 stem-tDnaK VLPs. The predicted structures of (**b**) the PR8 H1 stem and (**c**) tDnaK from *E. coli*. (**d**) Transmission electron microscope analysis of H1 stem VLP, H1 stem-tDnaK VLP, H1 VLP, or H1-tDnaK VLP. White bars represent 100 nm in the chimeric VLP images and 10 nm in the images of partial enlargement. The blue and grey bars above the right image indicate the lengths of the H1 stem domain and lipid layer. The cartoon of the H1 stem that was intercepted from the PR8 H1 structure (PDB: 1RU7) was generated using PyMOL software. The shape and size were compared by image overlay with TEM image and cartoon image. The height of the H1 stem in the cartoon measured by using PyMOL is approximately 10 nm.

Previous research indicates that tDnaK could activate the proinflammatory responses in host cells through TLR-4 recognition. It was reported that activating TLR-4 played an important role in the release of IL-1β, IL-6, and IL-10 cytokines from the activated monocytes [27]. To assess whether the recombinant tDnaK presented on the VLP membrane still retains the ability to activate cytokine responses in host cells, the human A549 cell line was used for the VLP treatment experiments. As shown in Figure 3, tDnaK-containing cVLP significantly enhanced the transcription levels of IL-1β and IL-6 genes in A549 cells. The caspase-1-dependent cytokine IL-1β exerts enhancement effects on the initiation of the protective Th1 and Th17 responses [28].

### 3.2. The cVLPs Induced Humoral and Cellular Immune Responses

We next assessed the immunogenicity of cVLPs in BALB/c mice. Mice were intraperitoneally immunized with 25 μg VLP immunogens adjuvanted with the incomplete Freund’s adjuvant per dose. The mock immunizations of mice was performed with PBS mixed with the incomplete Freund’s adjuvant as a negative control group. Two weeks after the last boost immunizations, the mice were bled to test for antibody titers of total IgG as well as the isotypes IgG1 and IgG2a specific to inactivated PR8 and NJ76 H1N1 viruses. All mice immunized with any type of these cVLPs significantly elicited H1N1-specific antibody responses of IgG, as well as the isotypes IgG1 and IgG2a (Figure 4). The H1-specific antibody titers elicited in both the H1 VLP and H1-tDnaK VLP immunization groups are comparable in magnitude, while the PR8 H1-specific rather than the NJ76 H1-specific antibody titers in both groups were stronger than those in the H1 stem VLP and H1 stem-tDnaK VLP immunization groups. Unexpectedly, we did not observe the enhancement effects of co-displayed tDnaK on H1-specific IgG titers. As the suboptimal H1-specific immune responses in all vaccination groups in this study, only in the H1-tDnaK VLP immunization group was viral neutralization of antibody titer detectable in microneutralization assays using both NJ76 and PR8 H1N1 viruses (Table 2).

Both the H1 stem VLP and H1 stem-tDnaK VLP immunization groups induced broad cellular responses and stimulated IFN-γ- and IL-4-secreting splenocyte populations (Figure 5a). The H1 stem-tDnaK VLP immunization group showed significantly higher levels of both IFN-γ- and IL-4-secreting splenocytes (Figure 5b). The strong splenocyte reactivation in the H1 stem VLP and H1 stem-tDnaK VLP immunization groups indicated the successful induction of vaccine-specific cellular immune responses. The mock-immunization group showed the baseline level of splenocyte activation.

### 3.3. Immunizations of the cVLPs Conferred Heterologous Influenza Protection against Lethal-Dose Infections

Prophylaxis potency was evaluated with mouse challenge studies. Three weeks after the last boost immunization, mice were intranasally infected with 3 × LD_50_ of mouse-adapted NJ76 H1N1 or PR8 H1N1 viruses. All mice immunized with H1-tDnaK VLPs (5/5) and H1 stem-tDnaK VLPs (5/5) survived the lethal dose challenge with NJ76 H1N1, while mice immunized with H1 VLPs (3/5) and H1 stem VLPs (3/5) were partially protected from death, indicating that the co-displayed tDnaK played important roles in enhancing vaccine protection efficacy (Figure 6a,b). Due to the suboptimal vaccine-induced immune responses, all the immunized mice experienced weight loss upon viral challenge. However, the mice immunized with cVLPs co-displaying tDnaK recovered faster than the mice immunized with VLPs displaying the antigen alone. Upon PR8 H1N1 infection, the H1-tDnaK VLP immunization group (4/5) had a slightly higher survival rate but similar weight loss compared with the H1 VLP immunization group (3/5) (Figure 6c,d). H1 stem VLP and H1 stem-tDnaK VLP immunization significantly reduced mouse lung viral titers on the 4th day post sublethal infection compared with the mock immunization (Figure 7). Notably, the H1 stem-tDnaK VLP immunization group had remarkably lower lung viral titers than other groups.

## 4. Discussion

Exploring appropriate vaccine platforms is significant to strengthening vaccine-induced specific immunity. It is well appreciated that the influenza VLP platform has remarkable enhancement effects on the immunogenicity of the presented antigens. The influenza VLP resembles the wild-type virion in many physicochemical characteristics; meanwhile, it also inherits excellent abilities to induce robust specific immunity. In this study, we designed and produced four types of influenza cVLPs, each displaying the full-length H1 or H1 stem on the membrane surface with or without co-displaying the tDnaK adjuvant protein. As expected, transmission electronic micrography revealed the expression and good presentation of the H1 stem antigens on the VLP membrane surface (Figure 2d). The tDnaK has biological functions of inducing cytokine responses in A549 cells in vitro and also exerts adjuvant effects on the protective immunity induction by H1 stem-tDnaK VLP immunization, as evidenced by the higher survival rate and remarkably lower lung viral titers of immunized mice after influenza virus infections. These results indicate that the VLP platform is an effective way of expressing the H1 stem antigen and suggest that the cVLP is a feasible vaccine vector for developing HA stem-based universal influenza vaccines. 

Stabilizing the HA stem in the trimeric state and pre-fusion conformation favors the formation of native antigenicity. In previous research, the trimeric state of the recombinant HA stem was stabilized by fusing an exogenous trimerization motif to the HA stem polypeptide [12,23,29,30]. However, in our study, we found that the intact H1 transmembrane domain could assist in immobilizing H1 stem trimerization on the VLP surface; on the other hand, the intact fusion peptide limits the H1 stem to the pre-fusion state. It has been reported that a variety of protective cross-reactive monoclonal antibodies target the HA stem in the pre-fusion state [8,9,31]. Therefore, such improvements in the H1 stem in our VLPs potentially enhance the protective antibody responses. Among the mechanisms by which monoclonal antibodies against the HA stem impede influenza virus replication typically involve the inhibition of the low-pH-induced HA conformational change, antibody-dependent phagocytosis, and antibody-dependent cellular cytotoxicity.

Specific T lymphocyte responses are critical for immune protection by the clearance of infected host cells [32,33]. Our previous vaccination experiments demonstrated that the IFN-γ secreting splenocytes from H1 stalk antigen-immunized mice were significantly recalled after restimulation with diverse peptides from H1 [26]. Similarly, this study showed that the cVLPs presenting the H1 stem also induced influenza H1N1-specific cellular immune responses. However, the in vivo protective levels of the cellular immunity specific for each particular T cell epitope still need to be further investigated. It was reported that immunizations with universal T cell epitopes-based DNA vaccines protected outbred mice from lethal-dose influenza infections. Meanwhile, another study implied that the cellular immune responses induced by immunizations with the recombinant HA2 polypeptide from the H3 subtype seemed not associated with immuno-protection [34].

The heat-shock protein chaperone DnaK has great potential as an adjuvant protein for vaccine development. The recombinant DnaK as a Toll-like receptor 4 ligand induced the phenotypic maturation of the murine bone marrow-derived dendritic cells and activated the production of proinflammatory cytokines in a dose-dependent manner [35]. Further vaccination studies showed the effective adjuvant effects of DnaK or its C-terminal fragment on the enhancement of immune protection [20,36]. Similarly, we also observed that mouse immunization with cVLPs co-displaying tDnaK provided better immune protection against lethal-dose influenza infection than immunization with VLP displaying the H1 or H1 stem alone. In our research, DnaK-specific antibodies would in principle form complexes with the cVLP-displayed C-terminal fragment of DnaK, potentially favoring the transport and availability of antigen in the lymphoid organs through follicular dendritic cells. In addition, DnaK-specific CD4^+^ follicular T helper cells could also interact with those activated B cells presenting specific peptides and subsequently provide pro-survival signals. Together, we reason that, in our research, the C-terminal fragment of DnaK is an attractive immune adjuvant. 

## Figures and Tables

**Figure 3 viruses-14-02109-f003:**
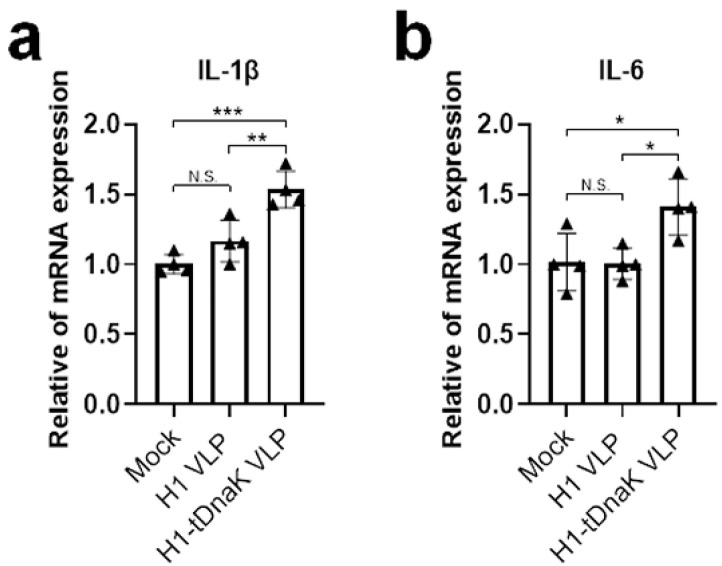
Analysis of the biological activity of membrane-anchored C-terminal domain of DnaK in vitro. Human lung carcinoma cell line A549 cell cultures were mock-treated with PBS or treated with NC99 H1 VLPs and NC99 H1-tDnaK VLPs, respectively. Cell cultures were collected 24 h after treatment for transcription analysis, and then total RNA from cell samples was extracted and reverse-transcribed into cDNA immediately. The gene transcription levels of (**a**) IL-1β and (**b**) IL-6 were quantified by qPCR. The results were presented as means with standard deviation (*n* = 4 per group). Each triangle symbol represents the relative value of cytokine mRNA expression level of each VLP-treated or mock-treated cell culture sample. Statistical significance was analyzed by *t*-test; *, **, and *** represent *p* < 0.05, *p* < 0.01, and *p* < 0.001, respectively. N.S. indicates no significance.

**Figure 4 viruses-14-02109-f004:**
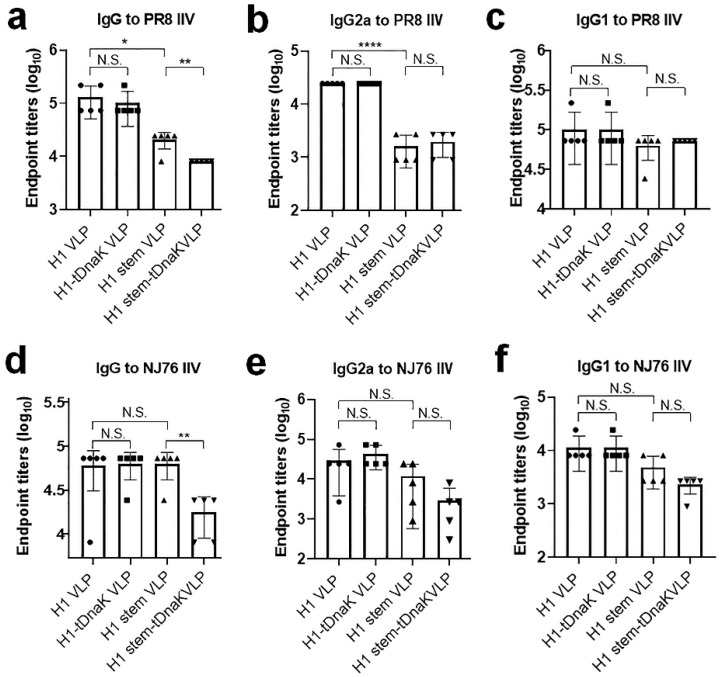
Immunization of mice with the chimeric VLPs’ results in specific serum IgG responses. The error bars represent standard deviations. (**a**–**c**) PR8-specific and (**d**–**f**) NJ76-specific serum antibody titers of (**a**,**d**) total IgG, (**b**,**e**) IgG2a, and (**c**,**f**) IgG1 were determined by using ELISA. Antibody endpoint titer is defined as the reciprocal of the highest dilution of a serum that gives a value of OD (450 nm) two-fold above the value of the pre-immune serum negative control. The results are presented as means with standard deviation (*n* = 5 per group). Each symbol represents the serum antibody endpoint titer of each mouse. Statistical significance was analyzed by *t*-test, *, **, and **** represent *p* < 0.05, *p* < 0.01, and *p* < 0.001, respectively. N.S. indicates no significance.

**Figure 5 viruses-14-02109-f005:**
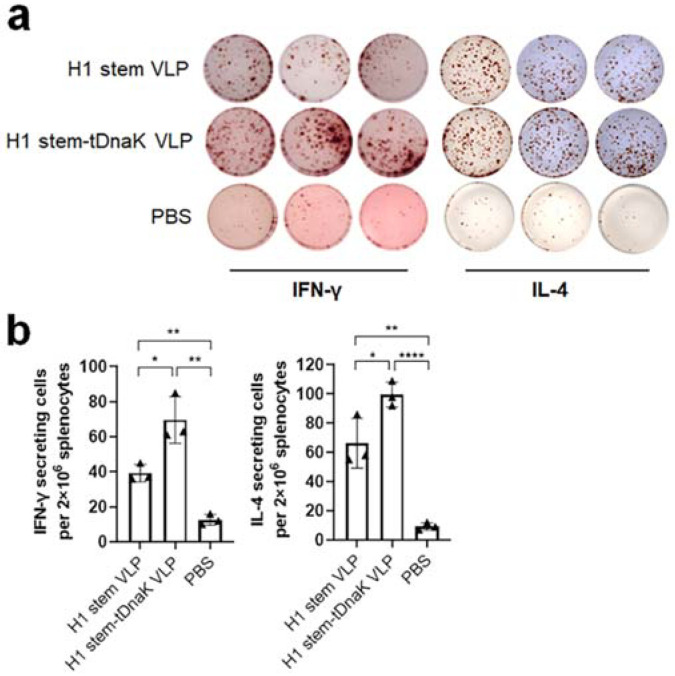
ELISPOT analysis of cellular immune responses. Mouse splenocyte samples (*n* = 3 per group) were collected on day 4 post sublethal-dose infections with 0.2 × LD_50_ of the NJ76 H1N1 virus. (**a**) Inactivated NJ76 H1N1 influenza virus re-stimulated IFN-γ- and IL-4-secreting cells were determined using ELISPOT assays. (**b**) The spot numbers of inactivated H1N1 virus-restimulated groups were quantified and compared in bar charts. The results were presented as means ± standard deviation. Each symbol represents the average spot number of the parallel samples from each mouse. The statistical significance was analyzed by *t*-test. *, **, and **** represent *p* < 0.05, *p* < 0.01, and *p* < 0.0001, respectively. N.S. indicates no significance.

**Figure 6 viruses-14-02109-f006:**
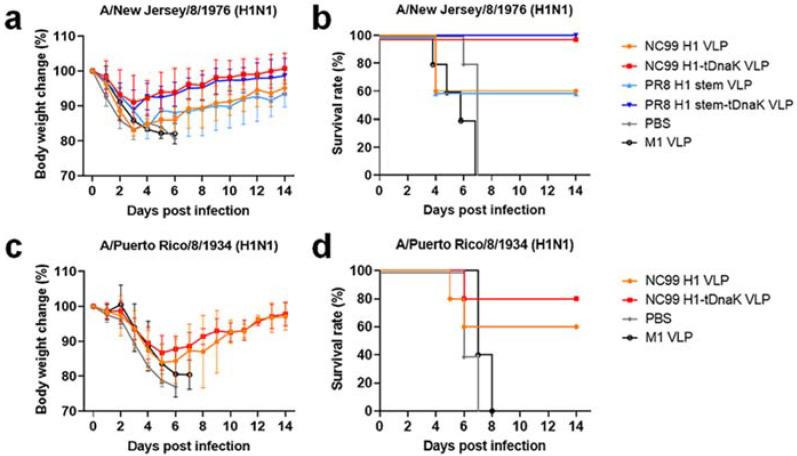
VLP protective efficacy in BALB/c mice. Body weight changes (**a**,**c**) and survival rates (**b**,**d**) of the immunized mice (*n* = 5 per group) upon challenges with 3 × LD_50_ lethal dose infections with NJ76 (**a**,**b**) or PR8 H1N1 (**c**,**d**).

**Figure 7 viruses-14-02109-f007:**
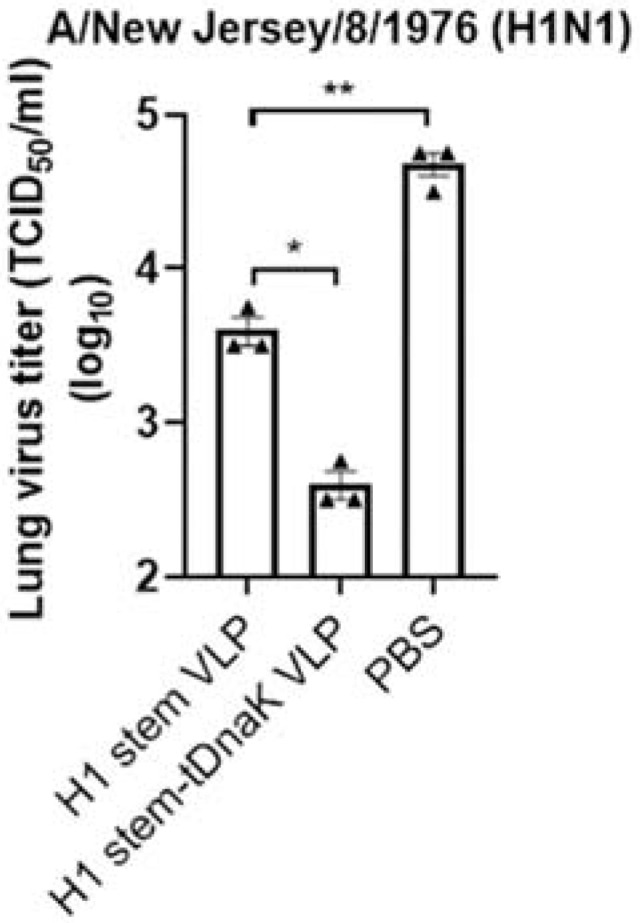
The determination of lung viral load after the sublethal infection. The purpose of the infection experiment is to compare the in vivo protectiveness among vaccination groups; the sublethal dose of mouse-adapted NJ76 was employed for infection for the consideration of lab-animal welfare. Mouse lung samples (*n* = 3 per group) were collected on day 4 post sublethal-dose infections with 1 × LD_50_ NJ76 H1N1 virus. Viral titers are expressed as TCID_50_/mL. The bars present the mean value ± standard deviation. Each symbol represents the average virus titer of the parallel samples from each mouse. Statistical significance was analyzed by *t*-test. * represents *p* < 0.05, and ** represents *p* < 0.01.

**Table 1 viruses-14-02109-t001:** Primers used in real-time quantitative PCR. F and R represent forward primer and reverse primer.

Primers	Sequences (5′-3′)
IL-1β-F	AGCTGATGGCCCTAAACAGA
IL-1β-R	TGGTGGTCGGAGATTCGTAG
IL-6-F	CCACTCACCTCTTCAGAACG
IL-6-R	CATCTTTGGAAGGTTCAGGTTG
GAPDH-F	CTCCTCCTGTTCGACAGTCA
GAPDH-R	CGACCAAATCCGTTGACTCC

**Table 2 viruses-14-02109-t002:** Virus neutralization titers determination. The highest dilution factors of the diluted heat-inactivated serum that were able to neutralize 150 × TCID_50_ of NJ76 or PR8 H1N1 virus are shown in the table.

Serum Samples	Virus Subtypes	
	NJ76 H1N1	PR8 H1N1
NC99 H1 VLP	<2	<2
NC99 H1-tDnaK VLP	4	8
PR8 H1 stem VLP	<2	<2
PR8 H1 stem-tDnaK VLP	<2	<2
PBS	<2	<2

## Data Availability

The authors declare that the data supporting the findings of this study are available within the figures and tables. All data are available from the corresponding author upon reasonable request.

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
