# Peer review of "Chimeric Virus-like Particles Co-Displaying Hemagglutinin Stem and the C-Terminal Fragment of DnaK Confer Heterologous Influenza Protection in Mice"

_viruses, 2022, doi:10.3390/v14102109_

Round 1
Reviewer 1 Report
The authors present work on using influenza virus-like particles as a vaccine platform.
They co-expressed the influenza matrix protein M1 in combination with full length influenza HA and HA stem constructs along with DnaK. Baculovirus expression systems were used along with sucrose gradient to purify the VLPs. EM, western blots, ELISA, and animals studies were sued to study the immune response to the VLPS. The authors indicate that the inclusion of Dnak into VLPs increases the cross-reactive immune responses to antigenically different HAs. The study is interesting but would be strengthen by some additions revisions:
There is general over hyperbolic use of certain words:
Abstract: remove “abundant” “well” and “efficient”
Introduction:
Lines 93-101 “ therefore,…..immunity” This should be in discussion.
Methods:
Please review the two current protocols suggested below and make sure methods and nomenclature meet the standard wording in the field.
A good review of standard influenza protocols is the following: Cuevas, F., et al. An In Vitro Microneutralization Assay for Influenza Virus Serology
Current protocols, 2022, https://doi.org/10.1002/cpz1.465.
Also, please read “Influenza, Current Protocols in Immunology (2001) 19.11.1-19.11.32”
Some portions of the methods sections are confusing. 2.8 Determination of viral load.
Usually viral load is determined as TCID50 or copies of vRNA/ml etc for mouse lungs and not HA titers. HA titers tells the amount of virus but not the amount of infectious virus. Please check this.
For example Line 229 “the presence of virus in the supernatant was assayed by measuring the hemagglutination activity in the supernatant and using the method of Reed and Muench for calculation.”
This should read “the presence of virus in the supernatant was assayed by measuring the hemagglutination activity in the supernatant and virus titers were calculated by TCID50 using the method of Reed and Muench for calculation.”
For Sf9 cells expression the volume of the cultures should be given. Was it ml or L?
In order to biochemically judge if VLPs were made a SDS gel of one of the sucrose gradients should be shown along with a corresponding western to show M1 and HA. If VLPs were made they should be a the bottom of the gradient.
Also, where are SDS-Page gels of the purified VLPs so the purity of the VLPs can be judged.
Figure. 2 Concerning EM, an additional panel of the spike be should added without the coordinate overlapping it.
Also, only one VLP is shown in figure 2d. A montage of VLPs should be shown to really show that the proteins are not free proteins but in VLPs. A montage of 4 different VLPs with help this.
Also, In EM methods, what was the camera used to collect the images? CCD? Film?
Can enough VLP images be collected to obtain 2D classes of the spikes. 2D classes with be more convincing than just one raw image shown in figure 2D. The spikes might be of different sizes and not confirm to the blue coordinates shown. This would be important to know.
For figure 2b. Why were these specific mutations used? Where they previously reported to stabilize the prefusion stem? Give references if appropriate.
For figure 3. What are the statistics for the other groups? e.g. mock vs H1 VLP, mock vs H1-tDnak VLP.
Through the manuscript hyperbole is used a bit. Such words as “robust” Line 328. Remove these words.
Figure 4. Dilutions curves are nice, but from the dilution curves End Point Titers or Aea Under the Curve (AUC) should be calculated to compare the immune response the different VLPs.
Review the aforementioned current protocols.
Table 1. Did the authors design these primers or are they commercially available? The authors should state in the methods why were these primers chosen. Are they standard primers used in the field?
Table 2. Is confusing. What does “4” actually mean. Is this from a Log scale.
Check the current protocols mentioned previously and report as standard in the field.
Line 367. The use of “heterologous” is confusing. “pan H1” is a better word for this study.
Figure 6. In text give actual survival numbers, e.g. 0/5, 5/5 mice survived challenge.
Figure 7. Can authors explain in the text why “sublethal infection” was used.
Line 403 “abundant expression” this is hyperbole, the authors don’t show a SDS-Page or give mg/ml of VLPs purified to justify “abundant”
Reviewer 2 Report
This article is very interesting and useful in the field of creating universal influenza vaccines on the base of virus-like particles (VLPs). The authors found original ideas to construct virus-like particles containing chimeric proteins and applied modern techniques to find out the effect of these VLPs in cells and mice. The study is carried out on high professional level. The experiments were well designed and performed. The results were correctly analyzed and discussed. Well done figures and tables perfectly illustrate and explain obtained data.
The appropriateness of definite components included in chimeric VLPs as also their safety for animals and humans needs additional research (these are future plans) and is a reason for discussion by readers and for the search of new creative approaches in this science direction.
However, there are some remarks and questions.
Section 3.1, Figure 1c. What does it mean p10 and pH?
Figure 4. What do figures 4a-4f mean? I did not find explanation neither in text (lines 329-343) nor in picture caption (line 346).
Section 3.3. Lines 369-388. Question on the base of Figure 6 (c, d).
Did you carry out experiments when mice immunized with PR8 H1 stem VLP or PR8 H1 stem-tDnaK VLP were challenged with A/PR8/1934 (H1N1)? If it was done, what kind results were obtained? If it was not done, would you explain why not, please? It is enough some words in the text.
